🔓 | **Open Peer Review** | Ecology | Research Article

# Shrub encroachment alters microbial community composition and soil carbon and nitrogen cycling functional genes in northern peatlands

Jie Ao,[1,2] Xinyu Tang,[1,2] Zhenxin Li,[1,2,3] Zhanhui Tang[1,2]

**ABSTRACT**  Changes in vegetation, such as shrub encroachment in grassland and wetland ecosystems, significantly influence soil microbial communities and biogeochemical processes. However, the specific impact of shrub encroachment on peatland ecosystems remains poorly understood. This study used a "space-for-time" approach, collecting soil samples from three encroachment stages—uninvaded, shrub invasion, and shrub invasion expansion—at two depths (0–30 cm and 30–60 cm). Metagenomic sequencing was used to assess the microbial community composition and functional gene dynamics. Shrub encroachment significantly alters soil physicochemical properties, nutrient availability, and microbial communities. Alpha diversity of bacteria and fungi was influenced by shrub encroachment and depth, whereas beta diversity varied mainly with depth. Functional carbon fixation genes (*korA* and *pps*) increased during shrub encroachment, while methane oxidation (*hdrA2*) and carbon degradation genes (*GH31 and GH51*) decreased before increasing. In addition, functional genes linked to nitrogen cycling (*nifD*, *nifH*, *amoA*, and *amoC*) declined, indicating a reduction in nitrogen fixation and nitrification pathways. Correlation and Mantel tests revealed that the total soil carbon content was the primary driver of these functional changes. These findings highlight the dynamic microbial responses to shrub encroachment and offer insights into the soil carbon and nitrogen-cycling mechanisms in peatlands.

**IMPORTANCE** Shrub encroachment is transforming peatlands and altering their ecological and biogeochemical functions. This study provides critical insights into how shrub invasion affects microbial communities and functional genes responsible for carbon and nitrogen cycling in peatland soils. By revealing the underlying genetic mechanisms, this study enhances our understanding of the consequences of vegetation shifts on ecosystem processes. These findings are essential for predicting and managing peatland responses to environmental changes, helping to preserve their role as vital carbon and nutrient reservoirs.

**KEYWORDS**  peatland, shrub encroachment, microbial communities, carbon cycle functional genes, nitrogen cycle functional genes

Peatlands are transitional ecosystems between terrestrial and aquatic environments. They are characterized by persistently waterlogged surfaces or shallow standing water and are among the most ecologically valuable ecosystems on Earth (1). Most peatlands are found in the Arctic, sub-Arctic, temperate, and subtropical regions (usually at lower temperatures) of the Northern Hemisphere (2). Although northern peatlands cover less than 3% of the global land area, they store 450–100 Pg of carbon as persistent peat, representing approximately 1/3 of the global soil carbon pool (2–4). Low nutrient availability in northern peatlands (especially bogs) and the resulting difficulty in

**Peer Reviewer** Patricia E. Arancibia-Avila, Universidad del Bio-Bio, Chillan, Bio-Bio, Chile

Address correspondence to Zhenxin Li, lizx542@nenu.edu.cn, or Zhanhui Tang, tangzh789@nenu.edu.cn.

The authors declare no conflict of interest.

See the funding table on p. 17.

decomposing soil organic matter, combined with low temperatures, waterlogging, and anaerobic conditions, suppresses microbial decomposition, leading to considerable soil organic carbon accumulation (5). Consequently, the rate of soil carbon accumulation in peatlands is higher than that in other ecosystems (6). Peatlands contribute directly to biodiversity conservation by providing habitats for specialized species adapted to nutrient-poor, cold, and water-saturated conditions, and indirectly as major global carbon sinks that mitigate climate change, helping to preserve species diversity across broader ecosystems (7–9). In recent years, shrub encroachment into the peatland ecosystems has become increasingly common because of global warming, anthropogenic activities, and changes in precipitation patterns (10). For instance, a continuous increase in shrub cover has been documented over the past few decades in the high-latitude peatlands of North America and Siberia (11, 12), and this encroachment trend has been widely observed across regional scales (13). Such encroachment can alter plant community composition and diversity, as well as key ecosystem functions such as hydrological processes and biogeochemical cycling (14, 15). For example, an increase in shrub cover can lead to greater root exudation, which stimulates belowground microbial activity by elevating the labile carbon content in the soil (16). However, increased shrub cover may also raise the concentration of phenolic compounds in both litter and peat pore water, exerting cascading effects on the biogeochemical processes in peatland ecosystems (17). The formation and maintenance of peatlands require stable hydrological conditions over centennial to millennial timescales; however, their vegetation composition can undergo substantial changes over decadal periods (18). This makes peatland ecosystems particularly sensitive to external environmental changes and anthropogenic disturbance. Moreover, considering that shrub encroachment is a common disturbance in mire ecosystems, understanding its effects on peatland ecological processes is critically important.

Shrub encroachment significantly affects the composition of soil microorganisms in peatlands. Changes in microbial community composition reflect environmental adaptability. Different microbial compositions possess distinct functional characteristics that influence the functional genes related to carbon and nitrogen cycling during organic matter decomposition and nutrient transformation (19, 20). Shrub encroachment changes soil environmental conditions, such as rainfall interception and transpiration, resulting in soil drying and aeration, altered litter composition, and increased soil nutrient availability. In addition, shrubs affect the soil microenvironment through canopy shading, which hinders moss growth (21). These changes in microenvironmental conditions gradually alter the soil environment in ways that promote further shrub growth, thereby accelerating the establishment and expansion of shrubs (22). In peatland hummocks, vascular plants such as shrubs face less abiotic environmental pressure. In hummocks, the groundwater table is lower than that in the surrounding flat areas, reducing the risk of root hypoxia and resulting in drier soil with enhanced nutrient availability. Furthermore, hummocks in peatlands are located at the top of a well-developed humus layer, and their vegetation height, coverage, aboveground and belowground biomass, species richness, and diversity are significantly higher than those inside the humus layers and surrounding plains (12). Shrubs can accumulate and enrich large amounts of nutrients, such as organic carbon and nitrogen, in the surface soil through the metabolism of litter and roots, producing a "fertility island" effect (23, 24), resulting in a higher nutrient content in hummocks densely populated with shrubs (12). Shrub encroachment is significantly affected by environmental changes. As shrubs establish and grow, the positive relationship between these environmental changes and soil microorganisms plays a positive role in shrub invasion. Shrubs may affect the composition of soil microbial communities and nutrient cycling processes in peatland soils by altering the quantity and quality of nutrient input, decomposition, and nutrient accumulation (25). Guo et al. (2020) observed that different vegetation types caused variations in soil microbial diversity and functions. One of the primary reasons for the successful encroachment of shrubs is the associated change in the

microbial community structure. These changes affect crucial ecological processes such as carbon and nitrogen cycling and organic matter decomposition (26), ultimately leading to altered feedback mechanisms between soil microorganisms and shrubs. Current research suggests that invading plants have a minor impact on soil microbial structure in the short term. Long-term interactions may disrupt the balanced symbiotic relationship established between native plants and soil microorganisms, ultimately significantly affecting the soil microbial communities (27). In addition, the vertical soil profile is a highly heterogeneous environment with clear environmental gradients that significantly influence the assemblage of microorganisms (28). Previous studies have shown that the soil nutrient content in the vertical direction of peat bog soil is the main factor regulating microbial assemblage; however, in deep soils with low nutrient availability, minerals such as manganese (Mn), potassium (K), and sulfur (S) have a greater influence (29). This unique vertical-profile environment contributes to the evolution and maintenance of microbial community diversity. Simultaneously, interactive networks between microorganisms play a crucial role in influencing the microbial community structure and function (30). These interaction networks enhance the efficiency of microbial resource utilization and improve microorganism survival and reproduction, affecting the overall ecological function of the microbial community. Soil microorganisms are the key drivers of ecosystem functions in peatlands and play an essential role in organic carbon accumulation and greenhouse gas emissions (31).

Increasing evidence suggests that shrub encroachment in the peatlands may alter the abundance and activity of microbial functional genes involved in carbon and nitrogen cycling, potentially influencing key biogeochemical processes. Understanding the interactions between functional gene abundance and soil physicochemical properties is crucial for effective peatland management. Soil microorganisms play crucial roles in carbon cycling, including carbon fixation, degradation, and metabolism (methane production and oxidation) (32, 33). Global warming has caused a decline in groundwater levels and soil moisture content in peatlands. Consequently, the anaerobic soil environment in the peatlands worldwide has changed to an aerobic one. Soil carbon and nitrogen cycling are regulated by complex microbial processes. Key carbon fixation pathways vary with oxygen availability and include both anaerobic (e.g., WL and DC/4-HB) and aerobic (e.g., CBB and 3-HP) mechanisms (34). Microorganisms drive key nitrogen processes, such as nitrogen fixation, nitrification, denitrification, and ammonium oxidation, involving well-characterized functional genes (e.g., *nifH*, *amoA*, *nirK,* and *nosZ*) (35–38). These microbial functions respond sensitively to environmental changes. Zhang et al. (39) found that global warming increased the abundance of microorganisms with carbon-degrading functions, leading to a significant decline in the soil organic matter (SOM) content. Previous studies have also shown that the abundance of nitrogen-cycling genes is closely associated with shifts in vegetation and soil properties under environmental stress (40). However, the patterns of change in the abundance of key functional genes related to carbon and nitrogen cycling due to shrub encroachment on peatlands are unclear. Given the widespread encroachment of shrubs into ecosystems, understanding the response of soil microbial communities to shrub encroachment and its impact on the abundance of carbon- and nitrogen-cycling functional genes is crucial.

In this study, we focused on a representative peatland ecosystem in northeastern China to explore the influence of shrub encroachment on soil microbial communities and their associated ecosystem functions. Given that shrub encroachment may alter soil physicochemical properties and microbial activity, we hypothesized that the progression of encroachment would (i) lead to significant shifts in the composition of bacterial and fungal communities, (ii) affect the abundance and diversity of microbial functional genes involved in carbon and nitrogen cycling, and (iii) be driven by specific environmental variables such as soil pH and nutrient availability. To test these hypotheses, we integrated field vegetation surveys, soil physical and chemical analyses, microbial community profiling, and metagenomic sequencing across three stages of shrub encroachment. This approach aims to elucidate the mechanisms by which shrub encroachment impacts

biogeochemical cycling in peatlands and to provide insights into ecosystem management and conservation.

## MATERIALS AND METHODS

### Sampling site information and sample collection

This study was conducted in Jinchuan Swamp Wetland (Fig. 1a), locally known as Jinchuan Xidadianzi, located approximately 1 km west of Jinchuan Town (42°20 56′ N, 126°22 51′ E; average annual temperature: 3.2°C; average annual precipitation: 774 mm; within the upper reaches of the Huifa River Basin, which belongs to the western region of Changbai Mountain), with an area of approximately 85 hm$^2$ and an average elevation of approximately 614 m a.s.l (41). Its formation is related to the volcanic crater lake formed by surrounding volcanic eruptions, which have evolved over a long period into the present wetland. This wetland originally covered an area of over 100 ha; however, some of its eastern parts were converted into farmland in the 1970s because of agricultural development impacts, with some peat areas being exploited, forming a small pond (northeastern part, approximately 0.23 hm$^2$). Adjacent to the farmland on the eastern side of the wetland is a drainage ditch that adversely affects the groundwater level of the adjacent wetland. In 1991, the Jilin Longwan Provincial Nature Reserve was established, with fences and signs erected around Jinchuan Wetland. In 2003, the State Council approved the establishment of the Jilin Longwan National Nature Reserve, with the Longwan Conservation Bureau designating the Jinchuan Wetland as a core conservation area, establishing warning signs and publicity boards around the wetland, and conducting regular patrols to protect it. Consequently, effective protection has been provided to the Jinchuan Wetland, making it a wetland of significant research value.

The local soil comprised dark-brown, partially developed swamp and meadow soil. The surface soil of the Jinchuan peatland is silty loam, whereas the underlying surface of the peat layer is characterized by low hydraulic conductivity, high density, and grayish-gray clay or grayish-yellow subclay. The peatland in the study area has an undulating surface with numerous small depressions, and geomorphological heterogeneity is apparent. Human-induced disturbances, coupled with hydrological fluctuations and unique topography, have helped form bare moist soil, which is beneficial for shrub seed germination and the creation of diverse wetland habitats. Peatlands in the succession stage show characteristics of herbaceous, nutrient-poor peatlands, and the vegetation community has obvious spatial heterogeneity, thus forming the current landscape pattern of the peatland. During the non-invasive stage of shrubs in peatlands, *Carex schmidtii* was the dominant species in the herbaceous layer, along with *Thelypteris palustris*, *Viola arcuata*, *Phragmites australis*, and *Hypericum japonicum*. These plants grow at sites where the groundwater level is relatively high and the bases of the plants are often flooded. With rising elevation and lowering groundwater levels, combined with peat accumulation, the amount of sediment trapped by the plant roots increased. The establishment of sphagnum moss provides conditions for shrub development and contributes to peatland formation during encroachment. The shrub layer mainly comprised *Spiraea salicifolia*, *Salix myrtilloides*, and *Betula ovalifolia*, whereas the herbaceous layer included *Thelypteris palustris*, *Carex schmidtii*, and *Phragmites australis*. As the *Carex* tussocks develop and organic matter accumulates, the water level further decreases, which could also be accompanied by the impacts of climate change and human activities. The shrub community entered an expansion phase, which likely started before 2013. At this stage, the dominant shrub species included *Spiraea salicifolia*, *Betula ovalifolia*, and *Salix rosmarinifolia*. This expansion reflects a successional shift in response to long-term environmental changes, rather than a purely climate-driven invasion. The composition of the herbaceous layer remained similar to that at the invasion stage.

To identify different stages of shrub encroachment in peatlands, we interpreted high-resolution remote sensing images (2013–2020) combined with a comprehensive

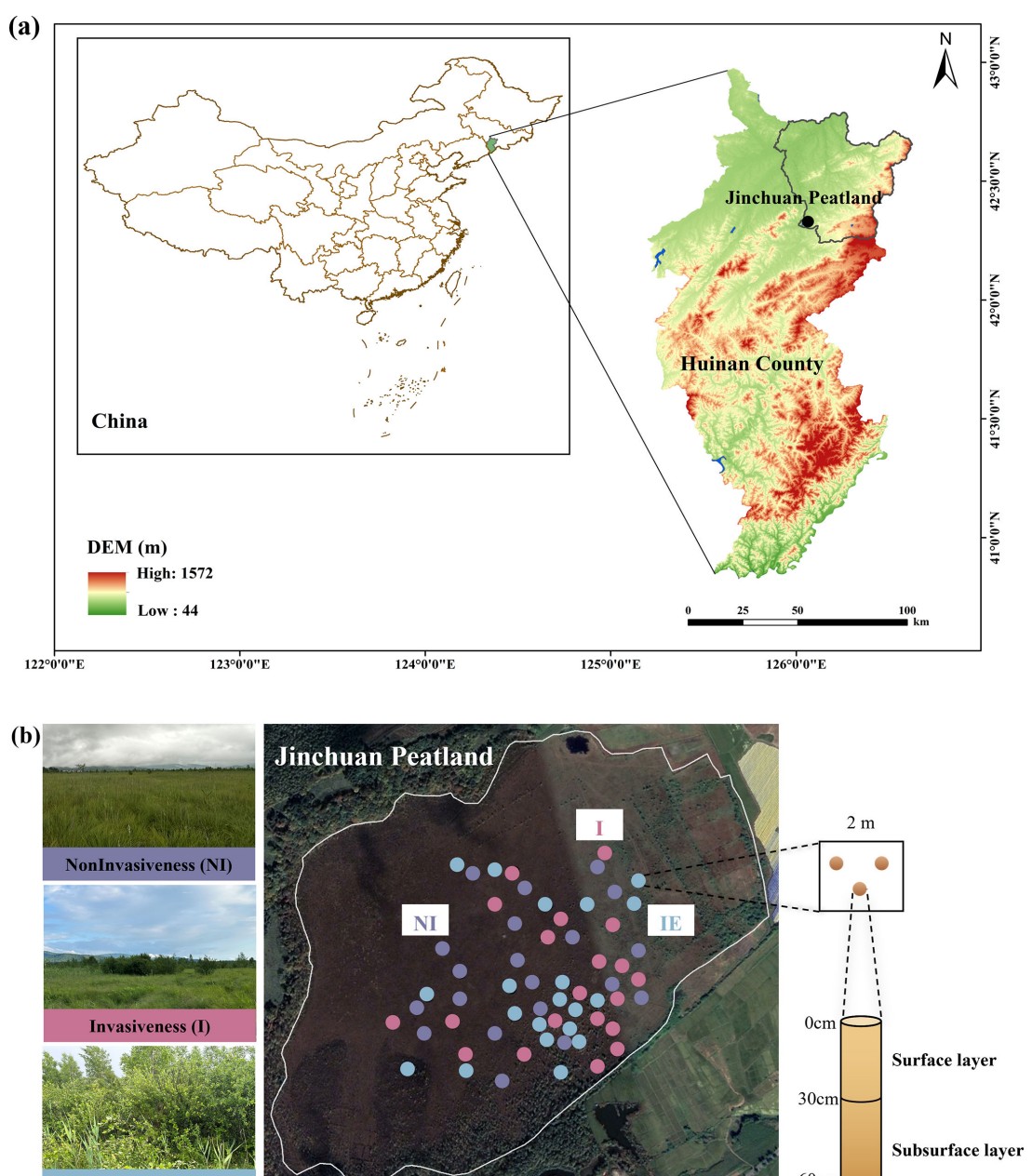

**FIG 1** (a) Study site location in Huinan County, China. The map was redrawn with ArcGIS based on the whole map of China and SRTM data of China, which was downloaded from the National Tibetan Plateau Center (http://https://data.tpdc.ac.cn/home); (b) photographs of the study site at each shrub invasion stage and sample collection. Photographs were downloaded from Google Earth Pro software based on the donation of sampling points' coordinates.

field vegetation survey conducted in 2022. The consistent climatic background of the region enables a space-for-time substitution approach. In total, 60 plots were selected in July 2023: 20 non-invaded (control), 20 shrub-invaded, and 20 shrub-expansion sites. Shrub encroachment in this peatland is likely the result of long-term hydrological alterations (e.g., artificial drainage channels) and climate-induced declines in the water table. Based on the average shrub basal diameter and remote sensing change history, we classified shrub invasion sites as having dominant shrubs with basal diameters between 0.1 and 0.7 cm (mostly established between 2013 and 2020), and shrub expansion sites as having basal diameters >0.7 cm (likely present before 2013). Thus, the observed encroachment represents both ongoing colonization and long-term expansion linked

to the historical degradation of peatlands. A peat drill was used to randomly select three points within a 2 × 2 m quadrat. After surface litter was removed, soil samples were collected from the soil profile at depths of 0–60 cm (divided into a surface layer: 0–30 cm and a subsurface layer 30–60 cm). A total of 360 soil samples were collected, representing three shrub invasion stages, with three replicates per site and at two depths (Fig. 1b). The collected soil samples were placed in sealed bags and stored at 0°C until transportation to the laboratory. The soil samples collected were divided into three parts: (i) for analysis of pH, soil water content (SWC), total phosphorus (TP), total carbon (TC), total nitrogen (TN), ammonia ($NH_4^+$–N), nitrate ($NO_3^-$–N), dissolved organic carbon (DOC), and total phenol (PC) (naturally dried at 25°C, then sieved through a 2 mm mesh); (ii) for determination of microbial biomass carbon (MBC) and microbial biomass nitrogen (MBN) (stored in a refrigerator at 4°C); and (iii) five soil samples were randomly selected from each of the three invasion stages and two depths, resulting in 30 soil samples then stored in an ultra-low temperature refrigerator at −80°C for microbial high-throughput sequencing analysis.

## Determination of soil fundamental physicochemical properties

Soil pH was determined using the potentiometric method (soil:water [wt/vol] ratio of 1:5). The SWC at different depths was determined using the drying method. The TP content in the soil was determined by digestion with $HF$–$HClO_4$, followed by molybdenum blue colorimetry (42). The TC and TN contents were analyzed using an elemental analyzer (EA3100, Euro Vector, Italy). The soil extract was filtered through a 0.45 µm filter membrane, and DOC was determined using a TOC analyzer (Shimadzu TOC-V CPN). Soil samples were extracted in 2 M KCl, and the concentrations of $NH_4^+$–N and $NO_3^-$–N were determined using a fully automated chemical analyzer (AMS SmartChem 200, France). The PC content was determined using spectrophotometry, and the MBN and MBC contents in the soil were determined using the chloroform fumigation extraction method (43).

## DNA extraction, metagenome sequencing, and annotation of DNA

Total genomic DNA was extracted from the soil samples using a Mag-Bind Soil DNA Kit (Omega Bio-tek, Norcross, GA, USA). The concentration and purity of the extracted DNA were determined using a TBS-380 and a NanoDrop2000, respectively. The quality of the DNA extracts was checked using a 1% agarose gel. The DNA extract was fragmented to an average size of approximately 400 bp using Covaris M220 (Gene Company Limited, China) for paired-end library construction. A paired-end library was constructed using NEXTFLEX Rapid DNA-Seq (BioScientific, Austin, TX, USA). Adapters containing the full complement of sequencing primer hybridization sites were ligated to the blunt ends of the fragments. Paired-end sequencing was performed by Majorbio Biopharm Technology Co., Ltd. (Shanghai, China) using an Illumina NovaSeq 6000 (Illumina Inc., San Diego, CA, USA) and NovaSeq 6000 S4 Reagent Kit v1.5 (300 cycles). A non-redundant gene catalog was constructed using CD-HIT (http://www.bioinformatics.org/cd-hit/, version 4.6.1) with 90% sequence identity and 90% coverage. High-quality reads were aligned to non-redundant gene catalogs to calculated gene abundance with 95% identity using Bowtie2 (https://bowtie-bio.sourceforge.net/bowtie2/index.shtml, version 2.5.4). Representative sequences of non-redundant gene catalogs were aligned to the NR database with an e-value cutoff of $1e^{-5}$ using Diamond (https://github.com/bbuchfink/diamond/releases, v2.1.13) for taxonomic annotation. Kyoto Encyclopedia of Genes and Genomes (KEGG) annotation was conducted using Diamond (https://github.com/bbuchfink/diamond/releases, v2.1.13) against the Kyoto Encyclopedia of Genes and Genomes database (https://www.genome.jp/kegg/) with an e-value cutoff of $1e^{-5}$. Carbohydrate-active enzyme annotation was conducted using hmmscan (software was obtained from https://github.com/EddyRivasLab/hmmer; documentation at https://www.ebi.ac.uk/Tools/hmmer/home) against the CAZy database (http://www.cazy.org/), with an e-value cutoff of $1e^{-5}$.

## Statistical analysis

Soil carbon and nitrogen cycle functional genes were analyzed by comparing all KEGG orthologues corresponding to carbon cycle functional genes (including carbon fixation, carbon degradation, and methane metabolism) with those of nitrogen cycle functional genes (including nitrogen fixation, denitrification, nitrification, nitrate reduction, and nitrite reduction). In this study, genes related to carbon and nitrogen cycles were identified through literature screening. The reads per kilobase per million mapped reads method was used to calculate gene abundance, eliminating the effects of gene length and sequencing depth to determine the abundance of functional genes. Bray–Curtis dissimilarity analysis (non-metric multidimensional scaling, NMDS) was conducted to represent the microbial community composition using the Meiji Cloud platform. The alpha diversity of soil microorganisms was evaluated using the Simpson and Shannon diversity indices. Soil physicochemical properties and microbial alpha diversity were analyzed at different soil depths and stages of shrub invasion using SPSS 26.0 (IBM SPSS Inc., Chicago, IL, USA) and ANOVA Duncan's test ($P < 0.05$). For data that did not conform to a normal distribution, non-parametric Mann–Whitney analysis (for different depths) and Kruskal-Wallis analysis (for different invasion stages) were used to compare significant differences between the different groups. Origin 2022 software was used to analyze variations in the relative abundance of microorganisms at different depths and stages of shrub invasion, as well as changes in functional genes related to carbon and nitrogen cycling after screening. The psych and pheatmap packages in R software (version 4.2.1) were used to visualize the correlation between soil carbon cycle functional gene abundance and environmental factors. The vegan package in R software was used to perform redundancy analysis (RDA) to evaluate the effects of environmental factors on functional genes related to carbon and nitrogen cycling. The Mantel test was used to analyze the relationship between soil physicochemical properties and the functional groups involved in carbon and nitrogen cycling. To differentiate between the two functional microbial groups, the abundance of functional genes associated with the carbon and nitrogen cycles was standardized using the following equation:

$$x' = \left[ \sum_{n=1}^{n} \left( x_i / \sum_{i=1}^{i} x_i \right) \right] / n (i = 1, 2, 3...; n = 1, 2, 3...),$$

where $x_i$ is the individual gene abundance of the samples, $i$ and $n$ indicate the number of samples and genes studied, respectively, and $x'$ is the normalized abundance of the C or N-cycling microbial groups.

## RESULTS

### Impact of shrub encroachment on peatland soil properties

The physical and chemical properties of the soil changed significantly ($P < 0.05$) due to shrub encroachment. During shrub encroachment, TC was significantly lower than during both the non-invasion and invasion expansion periods (Fig. 2a and e, $P < 0.05$). The TC content in the subsurface layer (350.44 ± 8.10 g/kg) was significantly higher than that in the surface layer (293.95 ± 25.40 g/kg) ($P < 0.05$). The concentrations of TN, $NH_4^+$–N, MBC, and $NO_3^-$–N during the invasion period were significantly higher than those during the non-invasion and invasion expansion stages (Fig. 2b through f, $P < 0.05$). In peatlands without shrub invasion, the TC, $NH_4^+$–N, and $NO_3^-$–N contents in the surface layer were lower than those in the subsurface layer, whereas the soil MBC content showed the opposite trend. The soil C:N:P stoichiometric ratio exhibited different trends during the different stages of shrub invasion. Soil C:N exhibited a trend of first decreasing and then increasing as encroachment developed, whereas the C:P ratio showed no significant change. The N:P ratio initially increased, followed by a decrease with the development of invasion (Fig. 2g through i). During the non-invasive stage, peatland soil had the lowest pH (4.83) and the highest water content (86.00%). The TP (0.77 g/kg) and

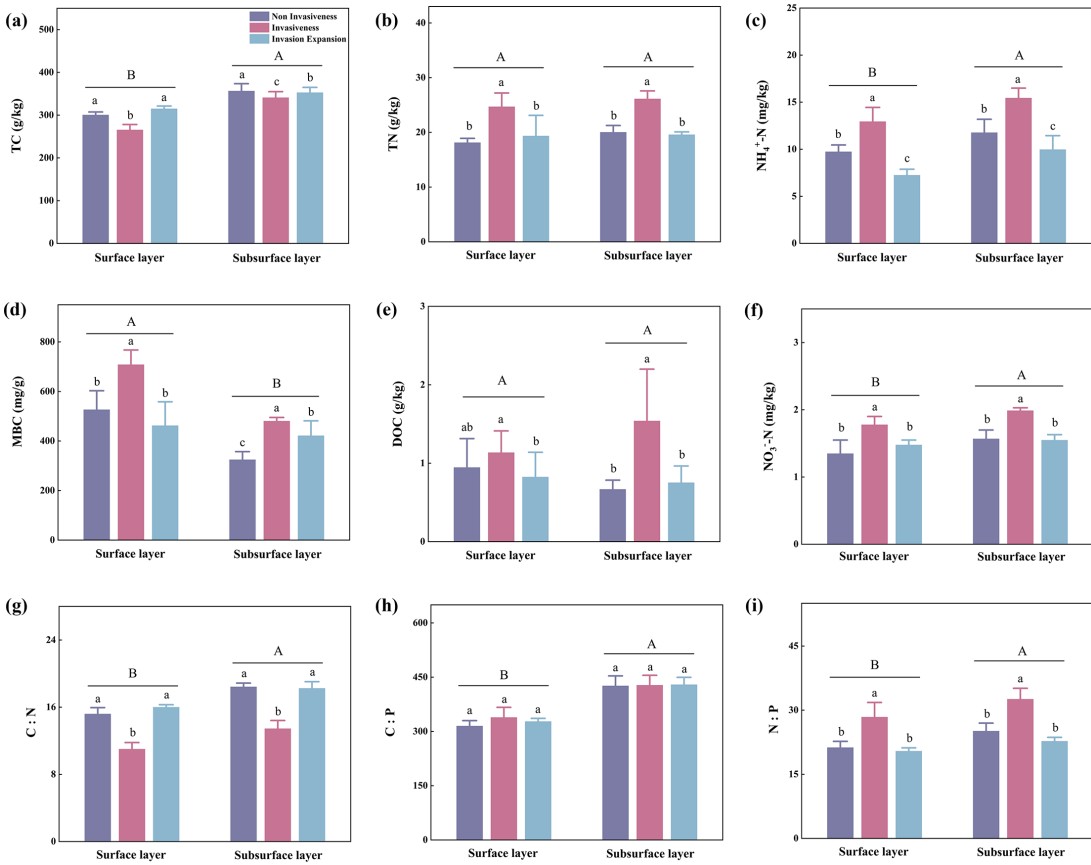

**FIG 2** Changes in soil physicochemical indicators (a–f) and soil C:N:P stoichiometric ratio (g–i) in peatlands at different shrub invasion stages and soil depths. Significant differences among different stages of invasion at the same depth are indicated by lowercase letters ($P < 0.05$), whereas differences at the same stage of invasion but at different depths are denoted by capital letters ($P < 0.05$).

PC (1.57 mg/g) levels were relatively low in the subsurface during the shrub invasion stage (Table 1).

## Impact of shrub encroachment on the composition and diversity of microbial communities

With shrub encroachment, significant variations were observed in the composition of the surface soil microbial communities (Fig. 3). In the surface soil layer, the relative abundance of Proteobacteria initially increased and then decreased with shrub invasion (average relative abundance: no invasion, 40.81%; invasion, 42.11%; and invasion expansion, 39.88%), whereas that of Chloroflexi initially decreased and then

**TABLE 1** Variation in pH, soil water content, total phosphorus, and total phenol across different stages of shrub invasion and soil depth[a]

| Stage | Soil depth (cm) | pH | SWC (%) | TP (g/kg) | PC (mg/g) |
|---|---|---|---|---|---|
| Non-invasiveness | 0–30 | 4.83 ± 0.10aA | 83.93 ± 1.51bA | 0.91 ± 0.07aA | 1.78 ± 0.11aA |
| | 30–60 | 5.00 ± 0.08aB | 86.00 ± 1.51bB | 0.81 ± 0.05abB | 1.61 ± 0.05aB |
| Invasiveness | 0–30 | 4.89 ± 0.07abA | 80.90 ± 1.36aA | 0.86 ± 0.09aA | 1.66 ± 0.07aA |
| | 30–60 | 4.97 ± 0.13aA | 83.97 ± 1.29aB | 0.77 ± 0.06bB | 1.57 ± 0.04aB |
| Invasion expansion | 0–30 | 4.97 ± 0.10bA | 82.01 ± 2.71abB | 0.95 ± 0.04aA | 1.74 ± 0.91aA |
| | 30–60 | 5.04 ± 0.06aA | 84.44 ± 1.09aB | 0.86 ± 0.04aB | 1.59 ± 0.13aB |

[a]TP, total phosphorus; SWC, soil water content; PC, total phenol content. Significant differences among different stages of invasion at the same depth are indicated by lowercase letters ($P < 0.05$), whereas significant differences at the same stage of invasion but at different depths are denoted by capital letters ($P < 0.05$).

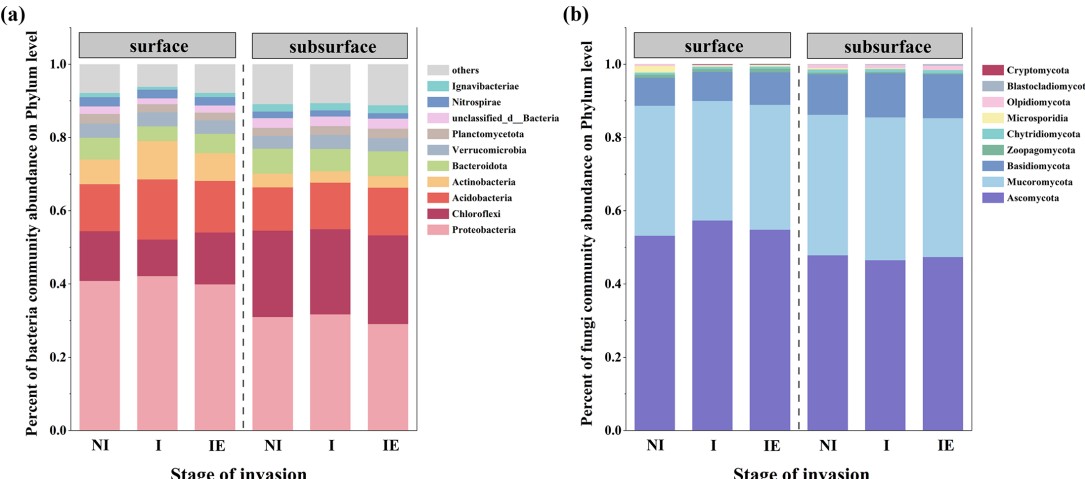

**FIG 3** Relative abundances of soil bacteria (a) and fungi (b) at different invasion stages and soil depths in peatlands.

increased (average relative abundance: no invasion, 13.6%; invasion, 10.01%; and invasion expansion, 14.18%). With increasing soil depth, the relative abundance of Proteobacteria gradually decreased (mean relative abundance: surface layer, 40.93%; subsurface layer, 30.57%). Ascomycota consistently maintained a high proportion of fungi at various stages of shrub invasion (mean relative abundance: non-invasive, 50.48%; invasive, 51.89%; invasion expansion, 51.09%). Blastocladiomycota was absent in the non-invasive phase of the surface soil but emerged during invasion and invasion expansion.

For alpha diversity, the Simpson index of surface soil bacteria in the shrub invasion stage was significantly higher than that in the non-invasive shrub and invasion expansion stages, whereas the Shannon index was significantly lower in the invasion stage than in the other two stages (Fig. 4a and b, $P < 0.05$). No significant changes were observed in the bacterial composition of the subsurface soil (Fig. 4a and b). In the surface soil, the Simpson index of fungi in the invasion stage exhibited higher values (Fig. 4c, $P < 0.05$), but the Shannon index showed no significant differences within the three invasion stages (Fig. 4d). In the subsurface soil, the Simpson index of fungi in the shrub invasion expansion stage was lower than those of the other two stages, and the Shannon index of fungi in the shrub invasion stage was lower than that of the invasion expansion stage, but it did not display a significant difference with the shrub no-invasion stage (Fig. 4c and d, $P < 0.05$). Furthermore, beta diversity analyses revealed that soil depth had a more pronounced effect on microbial community composition than invasion (Fig. 4e and f).

RDA and Mental tests were conducted to evaluate the impact of soil physicochemical properties on the composition of bacteria and fungi (including the top 10 species at the phylum level for bacteria and fungi), and Spearman correlation was conducted to elucidate the relationship between soil physicochemical properties and the diversity indices of bacteria and fungi (Fig. 5). According to the RDA results (Fig. 5a and b), the primary and secondary axes explained 94.01% of the total variation in the soil bacterial communities (RDA 1 85.5%; RDA 2 8.51%) and 92.55% of the total variation in the fungal communities (RDA 1 84.33%; RDA 2 8.22%). The TC content significantly affected bacterial community composition. The key factors affecting the structure of soil fungal communities were TC and MBC. The Mantel test results (Fig. 5c) indicate that the bacterial community was significantly influenced by TC (Mantel $r = 0.58$), C:P (Mantel $r = 0.57$), and MBC (Mantel $r = 0.32$). The TC and C:P ratios were the primary factors influencing alterations in the fungal community. Correlation analysis (Fig. 5d) revealed a close relationship between the microbial diversity indices and environmental variables, such as pH, SWC, TP, TC, and MBC.

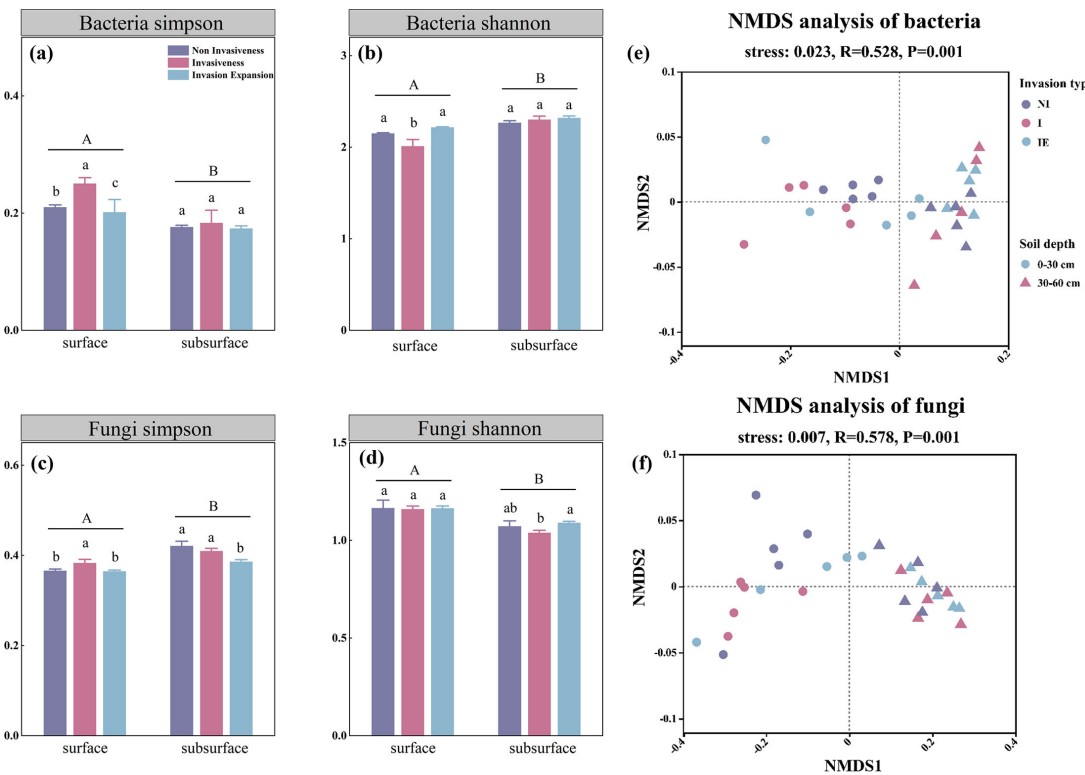

**FIG 4** Soil microbial alpha and beta diversities. (a and b) Differences in alpha diversity of peatland soil bacteria at different stages of encroachment and soil depths; (c and d) differences in alpha diversity of peatland soil fungi at different stages of encroachment and soil depths; (e) and f) NMDS analysis based on Bray–Curtis distance showing differences in beta diversity of soil bacterial and fungal communities.

## Response of functional gene abundance involved in the carbon and nitrogen cycling to shrub encroachment

A heat map analysis of the relational networks examined the relationship between soil physical and chemical factors and soil C:N:P stoichiometry against the abundance of C- and N-cycling genes (Fig. 6). In the C-cycling group, carbon fixation genes (*korA* and *pps*) showed significant positive correlations with TC and significant negative correlations with MBC ($P < 0.05$), besides *korA* genes showing significant positive correlations with pH, C:N, and C:P ratios. Methane metabolism genes (*acs* and *hdrA2*) had opposite correlations ($P < 0.05$) with SWC, pH, TC, C:P ratios, TP, and MBC. Carbon degradation genes, including *GH31*, *GH51*, and *GH74*, had similar correlations with the *korA* gene, and the soil TN and DOC content were significantly negatively correlated with the abundance of *GH74* ($P < 0.05$). Except for *hao*, the N-cycling group showed significant correlations opposite to those observed for the C-cycling genes. In addition, soil PC showed a significant positive correlation with *amoA*, which is involved in the nitrification process ($P < 0.05$). Soil pH was positively correlated with the abundance of certain C cycling genes (*korA*, *hdrA2*, and *GH51*) and negatively correlated with the abundance of nitrogen-fixing genes (*nifD*, *nifK*, and *nifH*), denitrifying genes (*narG*), nitrification genes (*amoA*, *amoB*, and *amoC*), ANRA (*nasA*), nitrogen mineralization genes (*GDH2*), and methane metabolism genes (*acs*). Soil $NH_4^+–N$ was significantly and negatively correlated with the abundance of N-cycling genes (*nifD*, *nifK*, *nifH*, *norB*, *amoA*, and *nrfA*) ($P < 0.05$).

Significant variations in the functional genes related to soil carbon and nitrogen cycling were observed at different stages of shrub invasion (Fig. 7). Among the functional genes of the C and N-cycling groups, *korA* presented the lowest gene abundance when the shrubs were not invasive. In contrast, most genes (*pps*, *hdrA2*, *GH31*, *GH51*, *GH74*,

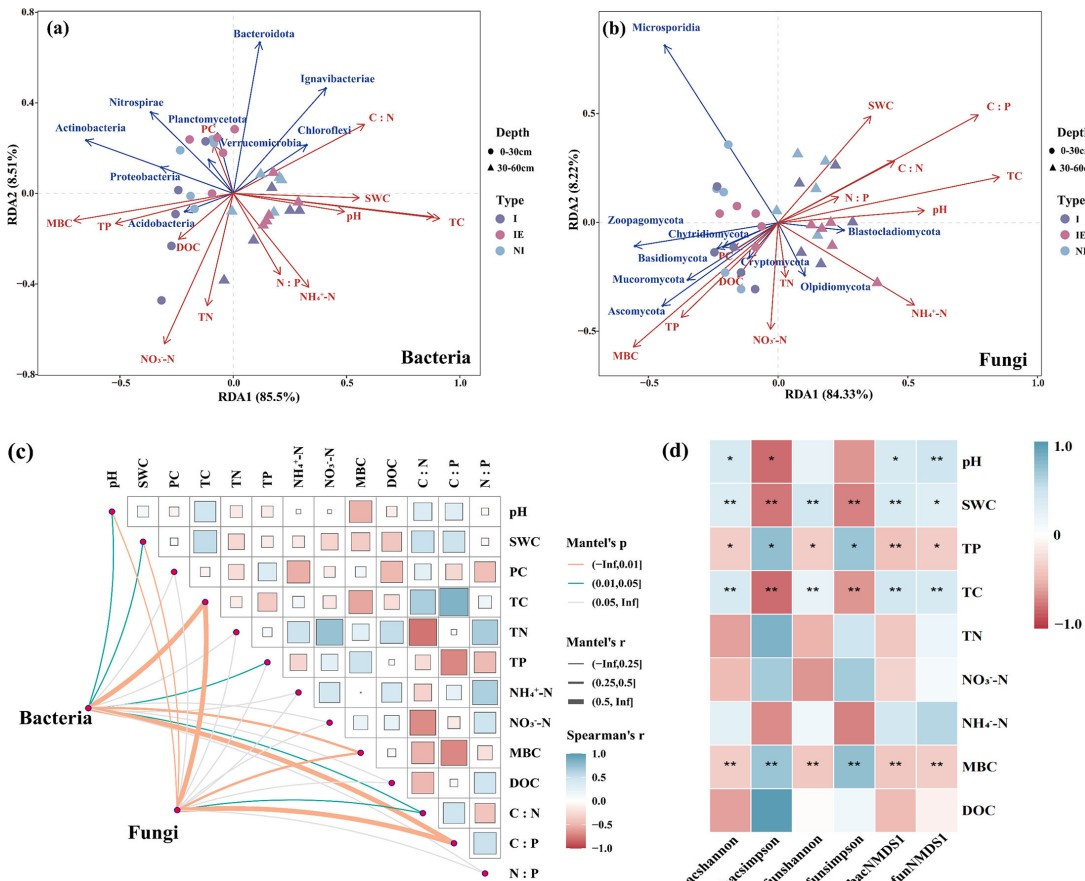

**FIG 5** Impacts of the soil physicochemical properties on the soil microbial communities (including the top 10 species at the phylum level for bacteria and fungi) by RDA (a and b). Mantel test between bacteria and fungi and soil physicochemical properties (c). Heatmap displayed the Spearman correlation between diversity indices and soil physicochemical properties; * indicates $P < 0.05$, and ** indicates $P < 0.01$ (d).

*nifD*, *nifH*, *narI*, *nrfA*, and *hao*) exhibited the lowest abundance in the presence of shrubs, and *acs*, *narG*, *amoA*, *amoC*, and *GDH2* had the lowest abundance during shrub invasion expansion. Among the C-cycle genes, carbon fixation genes (*korA* and *pps*) exhibited the highest relative abundance during the expansion stage of shrub invasion, whereas the two genes related to methanogenesis (*acs* and *hdrA2*) showed a large variation with shrub encroachment, and the summed abundance of genes for *acs* and *hdrA2* was the highest in the shrub invasion expansion stage but was lowest in the shrub invasion stage. In addition, carbon degradation genes (*GH31*, *GH51*, and *GH74*) exhibited relatively low abundance during the shrub invasion stage and increased again during the shrub expansion stage. Nitrogen-fixation genes, including *nifD* and *nifH*, exhibited higher gene abundance when the shrubs were not invaded and the lowest when the shrubs were encroached upon. The denitrification gene *narI* showed no differences between the shrub invasion and non-invasion treatments, whereas the abundance of *narG* genes showed a decreasing trend in the invasion expansion stage. For nitrification genes, the abundance of *amoA* and *amoC* was lowest at the shrub expansion stage, but the abundance of *amoA* increased and then decreased with shrub encroachment. The abundance of N-cycling genes (*nrfA*) did not differ between the shrub invasion and non-invasion stages, whereas the abundance of *hao* was higher in the shrub invasion expansion stage. The abundance of the nitrogen mineralization gene (*GDH2*) showed increasing and decreasing trends with shrub encroachment (Fig. 7; all $P < 0.05$).

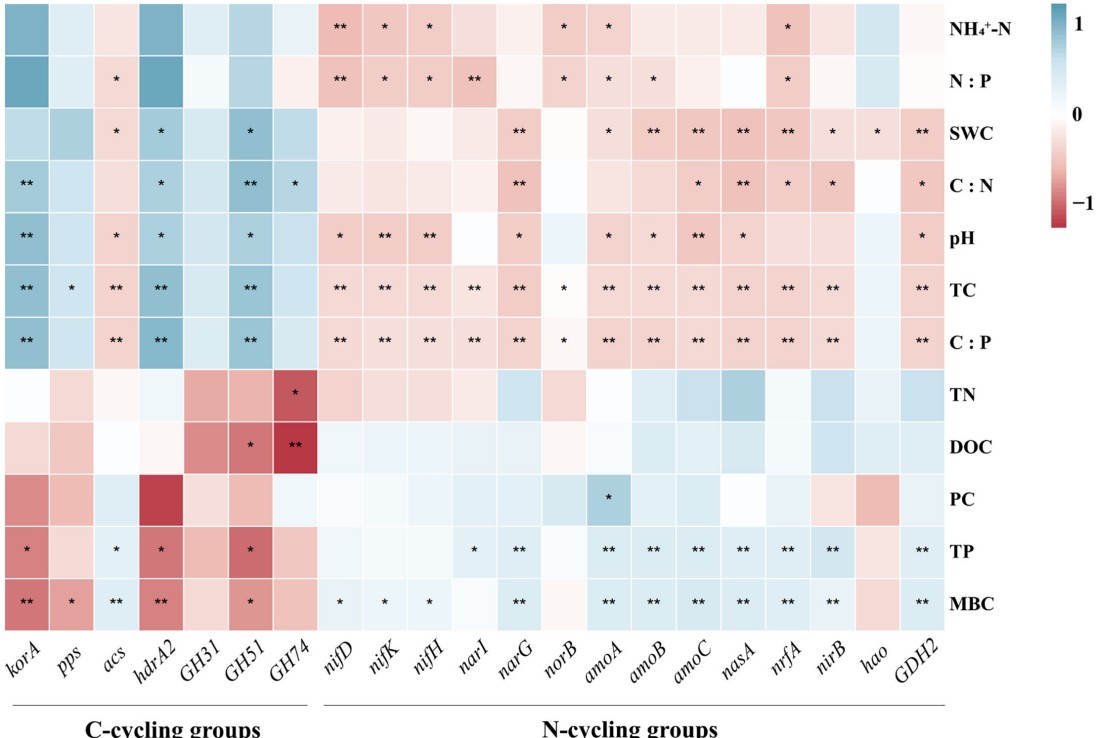

**FIG 6** Potential relationships between individual gene abundance and environmental factors. The red and blue colors show the negative and positive correlations, respectively, between the two variables. The deeper the color and the greater the asterisk, the stronger the correlation. Boxes without asterisks indicate no significant correlation.

## Correlation between microbial functional genes and soil physicochemical factors

Mantel's test was conducted to analyze the influence of soil characteristics and soil C:N:P stoichiometric ratio on the abundance of microbial CN cycle genes (Fig. 8). Mantel correlation analysis of specific CN cycling processes with environmental factors indicated a significant correlation between soil properties and microbial genes during shrub encroachment. Genes associated with the N cycle exhibited a stronger correlation with soil characteristics and soil C:N:P stoichiometric ratios than those associated with the C cycle. These analyses indicated that genes related to the C cycle were mainly influenced by TC, whereas N-cycle genes were significantly influenced by pH, SWC, TC, and C:P ratios.

## DISCUSSION

### Shrub encroachment and soil depth alter the structure of soil microbial communities

Shrub encroachment simultaneously influences plant litter input, soil properties, and microbial processes. Our study found significant differences in microbial alpha diversity (e.g., Simpson and Shannon indices) across shrub encroachment stages (Fig. 4, $P < 0.05$), indicating that vegetation changes may reshape microbial diversity and ecosystem function (20, 44, 45). Although the overall microbial community composition at the phylum level remained relatively stable, slight shifts in the relative abundances of dominant phyla such as Proteobacteria, Chloroflexi, and Acidobacteria were observed (Fig. 3). Our research demonstrated that during shrub invasion, the relative abundances of Proteobacteria and Acidobacteria in the soil were higher than those during the non-invasion and invasion expansion stages, whereas Chloroflexi exhibited the opposite

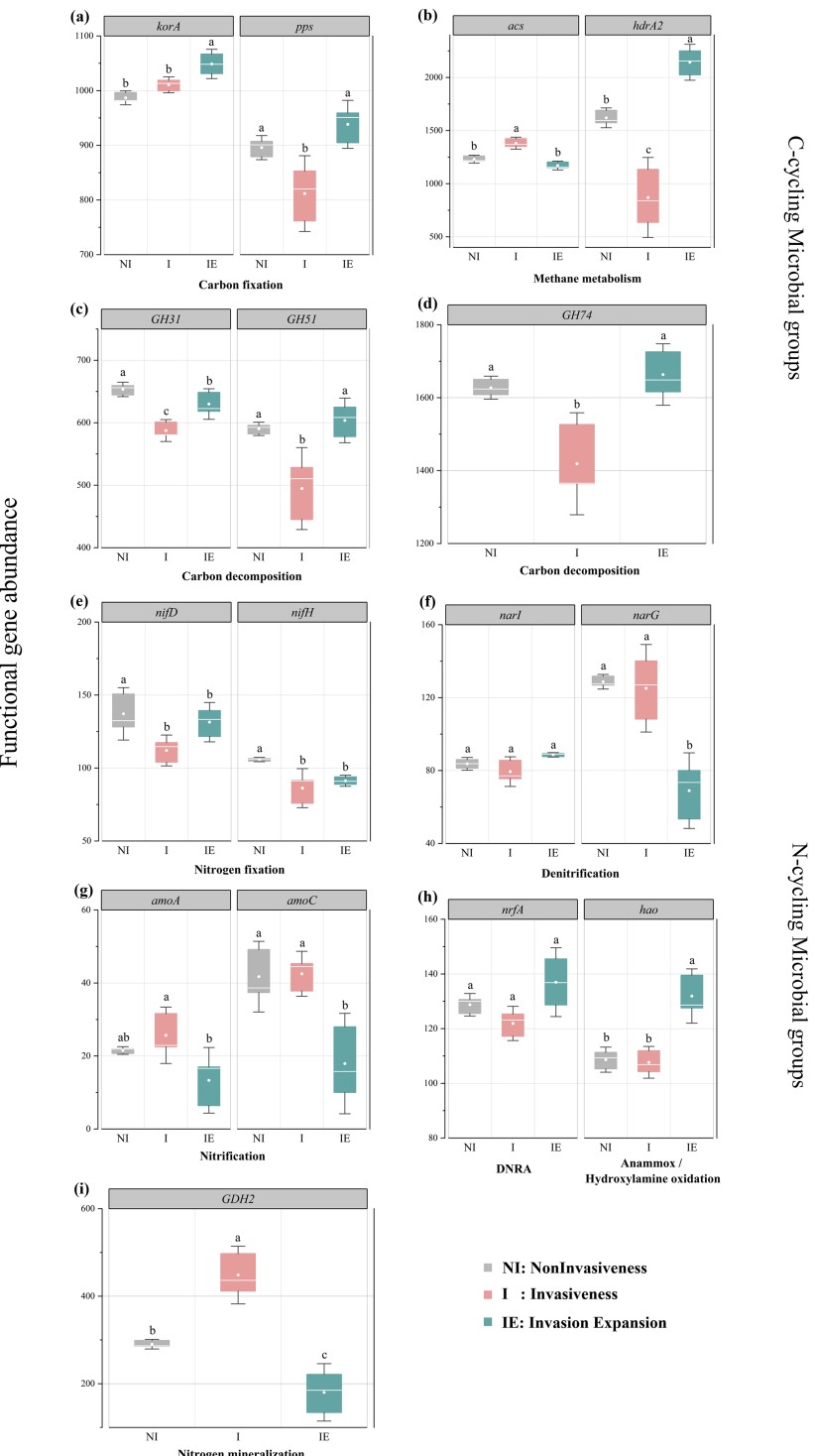

**FIG 7** Effects of non-invasiveness, invasiveness, and invasion expansion on the abundance of genes involved in carbon (a–d) and nitrogen cycling (e–i). The upper and lower boundaries of each box indicate the 75th and 25th percentiles, respectively, and the midline marks the median of the distribution of abundance values. Significant differences among different stages of invasion are indicated by lowercase letters ($P < 0.05$).

trend (Fig. 3). This phenomenon could be attributed to the initial stages of invasion when drying peatlands facilitated the encroachment of terrestrial plants, leading to a rapid increase in biomass, elevated evaporation rates, reduced SWC, and higher oxygen

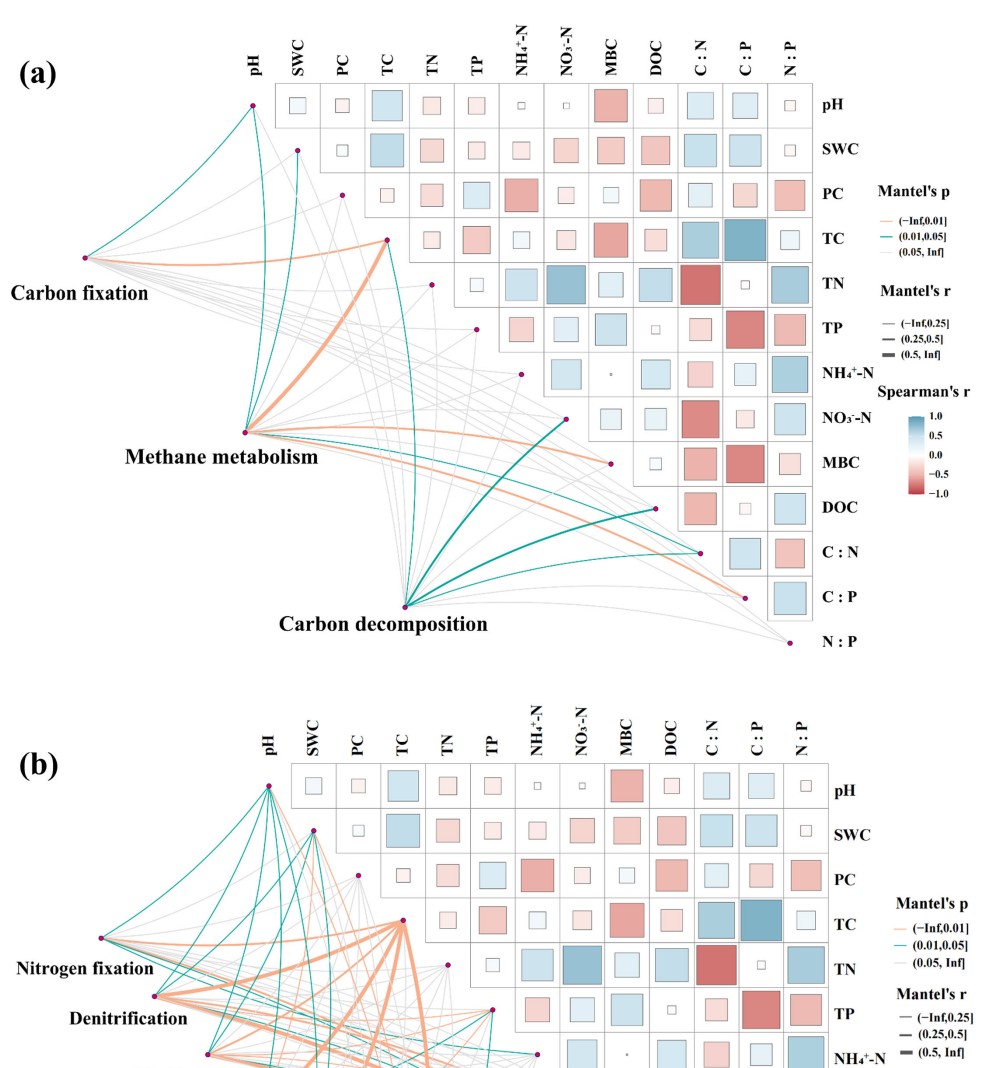

**FIG 8** Mantel's test for correlations between soil properties, soil C:N:P stoichiometric ratio, and carbon (a) and nitrogen (b) cycling gene composition. Color gradient representing Spearman correlation coefficient. The width of the edge reflects Mantel's *r* statistic, indicating the strength of the distance correlation, whereas the edge's color denotes the level of statistical significance.

levels. Field measurements confirmed this drying trend (Table 2), with average water levels of 4.0, 2.4, and 1.44 cm above the ground surface in the non-invaded, invaded, and expansion stages, respectively, indicating a progressive decline in soil moisture and water saturation across the encroachment gradient. Therefore, facultative anaerobes, such as Chloroflexi, are less prevalent in shrub-invaded soils. Previous studies have confirmed that a decrease in SWC increases soil oxygen levels and reduces the relative abundance of anaerobic microorganisms (46). When shrub invasion enters the expansion stage, organic matter input changes, and the depth and density of shrub roots

**TABLE 2** Changes in peatland water level at different shrub invasion stages[a]

| Stage | Water level (cm) |
| --- | --- |
| Non-invasiveness | 4.00 ± 1.00a |
| Invasiveness | 2.40 ± 0.42b |
| Invasion expansion | 1.44 ± 0.38c |

[a]Note: water level values represent the height (in centimeters) above the ground surface. Significant differences among different stages of invasion at the same depth are indicated by lowercase letters ($P < 0.05$).

increase, potentially improving the soil structure and nutrient content. Given that the phylum Chloroflexi is better at decomposing complex organic matter (47), especially with increased organic matter input, its metabolic activity and growth rate are accelerated, resulting in an increased relative abundance of Chloroflexi.

These functional shifts suggest that shrub encroachment may not dramatically alter the taxonomic structure of peatland microbial communities, but it affects their metabolic potential. For instance, during the shrub invasion stage, the lowest bacterial diversity (Fig. 4b) may be conducive to nitrate accumulation (48), consistent with the observed decrease in the soil C:N ratio and increase in nitrate concentration (Fig. 2c and f). In contrast, under prolonged shrub expansion and wetter conditions, ammonium usually accumulates due to suppressed nitrification and enhanced DON conversion (49). These dynamics indicate altered nitrogen cycling processes driven by microbial functional changes rather than shifts in dominant phyla. Soil moisture, pH, TC, and MBC were significantly correlated with microbial diversity indices (Fig. 5d), confirming their roles as key environmental drivers (50, 51). The pH also showed a significant positive correlation with bacterial diversity, consistent with previous research linking soil acidity to lower microbial diversity (52–54). Moreover, the relative abundance of bacteria and fungi was significantly associated with soil TC, C:P, and other nutrient ratios (Fig. 5c; $P < 0.01$), highlighting the role of stoichiometry in structuring microbial communities (55). Shrub litter and root exudates likely increase nutrient inputs and organic matter in invaded plots (56). However, the deep root systems of shrubs can lead to vertical nutrient redistribution, which may cause nutrient limitations and potentially reduce microbial diversity in deeper soil layers (57, 58). These mechanisms may explain the vertical trends observed in our study, where bacterial alpha diversity decreased with depth (Fig. 4). Our metagenomic data showed that, whereas the microbial community composition remained broadly similar, functional gene profiles were significantly reshaped along with the encroachment gradient.

## Potential impacts of shrub encroachment on key functional genes involved in soil carbon cycling

Variations in the abundance of soil microbial carbon cycling functional genes can influence carbon metabolism in microbial communities, affecting soil carbon cycling (59). In peatland ecosystems, the abundance of soil microbial carbon cycling genes is closely associated with vegetation type and soil properties (60, 61). Our study revealed that shrub encroachment significantly influenced the abundance of microbial functional genes associated with carbon cycling. Among the carbon fixation genes, *korA* showed significantly higher abundance during the invasion and expansion stages, suggesting increased activity of the energy-intensive 3-HP/4-HB pathway under improved nutrient and oxygen conditions (62–64). This was supported by Mantel tests showing that TC and pH were key drivers of gene variation (Fig. 8a), consistent with previous findings that higher pH promotes Calvin cycle gene abundance (65).

In the methane production pathway, *acs* and *hdrA2* exhibited stage-specific variations: *acs* peaked during the invasion stage, whereas *hdrA2* peaked during expansion (Fig. 7b), indicating a shift in methanogenic pathways with environmental changes (66). The research results suggest that the impact of shrub invasion on methane production (67) in northern peatland ecosystems is considerably complex (68). The

increase or decrease in methane production may be influenced by various environmental factors, such as soil organic carbon content, soil moisture, pH, and soil aeration conditions (69). For carbon degradation, the expression of genes such as *GH31*, *GH51*, and *GH74* initially decreased and then increased with shrub encroachment (Fig. 7c and d). This may reflect initial constraints due to low water content and complex shrub-derived litter inputs (70, 71), followed by microbial adaptation and enhanced decomposition during the expansion stage as nutrient demands rise (72, 73). Correlation analyses suggested that TC, $NO_3^-$–N, and C:N ratio were the major influencers of degradation gene abundance (Fig. 8a). Together, these findings indicate that shrub encroachment alters peatland carbon cycling through functional gene shifts, driven by changes in the water regime, SOM quality, and nutrient availability.

## Potential impacts of shrub encroachment on key functional genes involved in soil nitrogen cycling

Atmospheric nitrogen ($N_2$) can enter the soil-plant circulation system only through nitrogen fixation, which is affected by nitrogen-fixation genes (74). The relative abundance of these genes indicates the level of biological N-fixation activity (75). Shrub encroachment influenced the abundance of nitrogen fixation and transformation-related genes. The nitrogen-fixation genes *nifD* and *nifH* showed the lowest abundance during the shrub invasion stage but increased again during expansion (Fig. 7e), likely due to shifts in the dominant nitrogen-fixing bacteria (e.g., *Bradyrhizobium*) and improved organic substrate availability from increased litter and root exudates (76–79). Root-secreted compounds may also stabilize soil aggregates and promote organic nitrogen accumulation, further enhancing the abundance of nitrogen-fixing genes (80).

Nitrification genes (*amoA* and *amoC*) declined with shrub encroachment, possibly due to the increase in pH (Fig. 6), which suppresses ammonia-oxidizing bacteria (81, 82). After shrub invasion, the pH of peat soil increases, creating a mildly acidic environment that reduces microbial nitrification (49). This also explains the negative correlation between pH and the nitrification genes *amoA* and *amoC* (Fig. 6, $P < 0.05$), which is consistent with the findings of Xie (83). The denitrification gene *narG* was most abundant in the non-invaded stage and declined later, likely due to reduced soil moisture and higher SOM, both of which inhibit denitrifying microbial activity (84–88). These results suggest that early shrub invasion may promote nitrogen accumulation via enhanced microbial activity and gene expression, as reflected by higher TN, $NO_3^-$–N, and $NH_4^+$–N contents during the invasion stage (Fig. 2b and f) (89). In the expansion stage, genes associated with nitrate reduction and anammox pathways increased (Fig. 7h), which could reduce $NO_3^-$ leaching by enhancing $NO_2^-$ formation (90). Mantel analysis confirmed that TC was a major factor driving nitrogen cycling gene patterns (Fig. 8b), consistent with the role of organic carbon as an electron donor in nitrogen reduction (91).

This study revealed the effects of shrub encroachment on soil microbial carbon and nitrogen functional genes using metagenomic analysis. However, the intrinsic mechanisms linking microbial functions to environmental changes remain unclear. In addition, our use of 0–30 cm and 30–60 cm depth intervals may confound recent microbial responses with legacy effects from long-term peat accumulation. Future studies should apply approaches such as DNA-stable isotope probing and amino sugar tracing to clarify microbial-mediated C and N cycling processes. Simultaneous measurements of greenhouse gases (e.g., $CH_4$ and $CO_2$) are also needed to better assess the impact of shrub encroachment on ecosystem carbon dynamics.

## Conclusion

This study demonstrates that shrub encroachment alters peatland soil physicochemical properties, with TC initially decreasing and then increasing, whereas inorganic nitrogen ($NH_4^+$–N and $NO_3^-$–N) exhibited the opposite trend, both of which influence shrub

establishment. However, the study found that microbial community changes during shrub encroachment were mainly reflected in bacterial diversity, with limited shifts in the overall community composition. Specifically, bacterial alpha diversity was significantly affected by encroachment, whereas beta diversity was significantly influenced by changes in soil depth. Surface soil microbial composition shifted significantly with shrub invasion, whereas subsurface communities remained relatively stable. Functionally, shrub encroachment increased the abundance of carbon fixation genes (e.g., *korA* and *pps*), altered methanogenesis-related genes (*acs* and *hdrA2*), and modulated carbon degradation genes (*GH31*, *GH51*, and *GH74*) in a stage-dependent manner. Nitrogen cycling genes (*nifD*, *nifH*, *amoA*, and *amoC*) decreased with the shrub encroachment, suggesting weakened nitrogen fixation and nitrification. Mantel and correlation analyses showed that carbon- and nitrogen-cycling genes were significantly associated with TC, $NO_3^-$–N, and nutrient ratios (C:N, and C:P). These findings enhance our understanding of microbial functional responses to shrub encroachment and provide guidance for the conservation and management of peatlands.

## ACKNOWLEDGMENTS

This research was funded by the National Key R&D Program of China, grant numbers 2023YFF1304604-3, and funded by the Key R&D Program of Tibet Autonomous Region, China, grant numbers XZ202401ZY0110.

## AUTHOR AFFILIATIONS

[1]Key Laboratory of Wetland Ecology and Vegetation Restoration, Ministry of Ecology and Environment, Changchun, China
[2]Key Laboratory of Vegetation Ecology, Ministry of Education, Northeast Normal University, Changchun, China
[3]School of Ecology and Environment, Tibet University, Lhasa, China

## AUTHOR ORCIDs

Zhenxin Li http://orcid.org/0000-0001-6016-5972
Zhanhui Tang http://orcid.org/0000-0002-4530-2091

## FUNDING

| Funder | Grant(s) | Author(s) |
| --- | --- | --- |
| National Key Research and Development Program of China | 2023YFF1304604-3 | Zhenxin Li |
| Key R&D Program of Tibet Autonomous Region, China | XZ202401ZY0110 | Zhenxin Li |

## AUTHOR CONTRIBUTIONS

Jie Ao, Conceptualization, Formal analysis, Investigation, Methodology, Writing – original draft, Writing – review and editing | Xinyu Tang, Investigation, Methodology | Zhenxin Li, Conceptualization, Investigation, Methodology, Writing – review and editing | Zhanhui Tang, Conceptualization, Methodology, Supervision

## DATA AVAILABILITY

The raw sequencing data generated for this study have been deposited in the NCBI Sequence Read Archive (SRA) under BioProject accession number PRJNA1279354. All other relevant data supporting the findings of this study are available from the corresponding author upon reasonable request.

## ADDITIONAL FILES

The following material is available online.

### Supplemental Material

**Supplemental figure and tables (Spectrum00542-25-s0001.docx).** Data related to the metagenome.

### Open Peer Review

**PEER REVIEW HISTORY (review-history.pdf).** An accounting of the reviewer comments and feedback.

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
