## [Reviewer comments · Microbiology Spectrum]

Microbiology Spectrum

Shrub encroachment alters microbial community composition and soil carbon and nitrogen cycling functional genes in Northern peatlands

Jie Ao, Xinyu Tang, Zhenxin Li, and Zhanhui Tang

Corresponding Author(s): Zhenxin Li, Northeast Normal University School of Environment

Review Timeline:

Submission Date:	February 21, 2025
Editorial Decision:	May 5, 2025
Revision Received:	May 22, 2025
Accepted:	June 10, 2025

Editor: Katharina Kujala

Reviewer(s): Disclosure of reviewer identity is with reference to reviewer comments included in decision letter(s). The following individuals involved in review of your submission have agreed to reveal their identity: Patricia E Arancibia-Avila (Reviewer #2)

Transaction Report:

DOI: <https://doi.org/10.1128/spectrum.00542-25>

Re: Spectrum00542-25 (**Shrub encroachment alters microbial community composition and soil carbon and nitrogen cycling functional genes in Northern peatlands**)

Dear Dr. Zhenxin Li:

Thank you for the privilege of reviewing your work. Below you will find my comments, instructions from the Spectrum editorial office, and the reviewer comments.

As you will see, one of the reviewers has raised quite a lot of criticism and recommended rejection of the manuscript. As the second review was more positive, I will give you the possibility to modify your manuscript addressing the reviewers' concerns.

Revision Guidelines

Sincerely,
Katharina Kujala
Editor
Microbiology Spectrum

Reviewer #1 (Public repository details (Required)):

all metagenome data and related values need to be deposited as a general practice

Reviewer #1 (Comments for the Author):

The MS titled " Shrub encroachment alters microbial community composition and soil carbon and nitrogen cycling functional genes in Northern peatlands" seeks to evaluate the geochemical and microbial changes (metagenome) associable with shrub encroachment in a one peatland complex that has undergone degradation due to agricultural land use and later converted to protected area. This study selects multiple sites in peatland complex and organized in 3 types according to the shrub abundance or invasion. No hypothesis or novel questions are listed in the introduction, although a revision of shrub encroachment in many non-peatland ecosystems is provided. I have several concerns about the information used to structure this manuscript, lack of novel research goals besides measuring changes in response to shrub abundance, poor sampling design, and the lack of a more comprehensive analysis of metagenomes limits my enthusiasm for this work. I find that likely due to confounding effects of ecosystem degradation (along with shrub increase), pooling of soil samples, and shallow metagenome analysis, the findings are not novel, or establish new or unexpected findings, or new mechanisms, thus I do not recommend this article for publication in this journal (perhaps a more specialized one will be more suitable)

Below, I provided several components to substantiate my concerns.

The introduction's arguments about the drivers and effects of shrub encroachments are primarily built on semi-arid, dryland, or grassland literature (citations 6 to 16), wetlands, marshes, etc. Although there is nothing to prohibit using broader literature to frame a problem, my comments is that authors need to better revise the literature of peatlands and shrubs and shrub encroachment as in this example (<https://doi.org/10.1111/gcb.16904>) and synthesize their scientific problem using literature that hold facts on shrub encroachment on peatlands or explain that the peatland in study is somewhat more relatable to the ecosystems revised in introduction. Of the 95 citations an apparent ~18 are related to shrubs and peatlands or peatlands in general. Please reconsider revising shrub peatland literature and rearticulate lines 56-80. Also, I noticed the intro has sentences related to the main results of study. Results and discussion should be reserved for later and not necessarily sprayed at introduction, introduce the questions and hypothesis for study instead. Lines 130 to 150 are a long listing of genes and pathways, making this intro lengthy and not really clarifying the rationale for this study.

When trying to identify the process of shrub encroachment, the site description section does not explain the process of "shrub encroachment" and if such is a process of invading species or a successional stage in this system. L555 suggests that this is a shrub invasion: when did it start (from 70s?), however this is likely a combination of the effects of man-made changes (like channel) more than a process of invasion due to changing climate. This study may need to be reframed under a historical degradation process as a better way to frame the space-for- time gradient used here. From revising the site pictures and sample distribution it suggests that the "invasion expansion" is located primarily on the side affected by agriculture or land use thus a question suggesting the observations are likely the effects of the nearby in used land and one aspect is shrub expansion. Looking at the measurements and experimental design it is not clear how the effects of proximity to degraded land or shrub effects can be separated.

The soil sampling by two depths (0-30, and 30-60 cm) may also be challenging to truly sort the effects of vegetation changes on microbes. For instance, the annual accretion rate of OM in peatlands is in the range of some mm, which assuming a high rate of 1 mm/y (<https://doi.org/10.5194/cp-17-2633-2021>) then 30 cm will evaluate a mix of soils of 1- 300 years, and this sampling may be even or unevenly mixed since less than 1 gr is commonly used for data generation. The sampling design should have been restricted to a shallower depth to better capture the last decadal process. Also, I understood that 20 samples were taken for each category but NMDS n Figure 3 only shows 5 samples per category (sampled at 0-30 and 30-60, thus 10 samples) which will be a major undersampling.

The results section is primarily of descriptive nature with significance values or correlation scores provided. However, for functional genes section which is based on metagenomes, the depth of sampling, evenness of gene representation is not presented (can be a supplementary data comment) to justify reasonable comparisons of the frequency of gene detection. Also, I am surprised this study limits their metagenome analysis to a few functional markers instead of taking the opportunity to evaluate the pathways, types and other categories that can be associable to the large change in communities as proposed here. Perhaps metagenome data is not deep enough.

The discussion from L289 to 361, is lengthy argumentation of why some parameter goes up or down in the three categories evaluated here. For instance, the initial 3.5 pages of discussion do not show much of novel discovery beyond explaining why some of the geochemical observations align with a changing vegetation coverage which should be expected. And then the composition of bacterial communities remains similar across the tree categories, which does not surprised me given that 30 cm of mixed soils could be pooling recent and older soils and the vertical mixing in soil peatlands is not too high depending of the water table dynamics. Also arguments about Chloroflexi reducing in abundance due to reduced soil content and oxygen penetration is provided but water level records not provided to substantiate this explanation. Functional genes and correlations with environmental variables were also pursued, but such correlations do not show cause-effect or mechanisms, etc, hence the section bringing a limited learning value.

Below, find some specific comments to consider and the overall recommendation to make the discussion smaller, closer to the

data, and identify the most meaningful findings along with detailing the limitations of the study as pointed out in the paragraphs above.

L 47-48: in what sense peatlands protect global diversity? do they hold diversity themselves or by helping against climate change protect diversity overall. You may want to clarify and add a citation.

L57-58: is there a descriptor of the magnitude of this problem? like frequency, area, or documented trend in regions? Incorporating that will help recognize the magnitude of this process

L61: what do you mean by ecological security, and how the example of a sub-arid and alpine region are relevant to peatlands.

L63-64: is the statement of shrub invasion also observed in northern peatlands. Citations 8,9 and 10 are of grasslands and a Florida marsh gradient. Covering this literature on Northern wetlands and especially peatlands will be the best approach

L 69-70: what is the basis for this? and what is a long time (decades or centuries?); vegetation makeup can change significantly in a decadal timeframe see this example <https://www.sciencedirect.com/science/article/pii/S1470160X22012043>

L74-75: is a statement on shrub affecting microbes in peatlands. Is this meant to be a result or based on a citation of published work. neither citation 15 and 16 provide such evidence.

L92 has the concept of Fertility islands been observed in many peatlands beyond Tibet ones?

L 107: replace alien for "invading"

L123-124: The introduction is not the place to present results

L218: RDA? Introduce all your abbreviations

L323-325 How does this extreme sensitivity is manifested? For instance, what level of change in composition and activity would a change of 1 degree of annual mean temperature elicit? I would recommend avoiding aggrandizing statements.

L289- L522 : the discussion needs to be synthesized or summarized and better focus main points since, at this point, it reads as a lengthy section pointing to similar info than results plus a bunch of possibilities that can explain or agree with various and not necessarily cohesive observations.

In the conclusions section, L665-666 indicates that Bacteria were the main drivers of soil microbial community changes, but this is in contrast with the discussion section (361-362), stating " The composition of soil microbial communities remained similar during the three stages of shrub invasion".

L969-974 citation 80 and 81 is the same twice

Reviewer #2 (Comments for the Author):

The study's objective is clearly stated, addressing the impact of shrub encroachment on peatland ecosystems by analyzing microbial communities and functional genes related to carbon and nitrogen cycles.

The results effectively answer the research question by demonstrating significant shifts in microbial communities and functional gene expression, highlighting soil carbon as the primary driver of these biogeochemical changes.

It is necessary to explain the abbreviations in the table title, even if they are already defined in the text. This will ensure clarity for readers who may refer directly to the table without revisiting the main text.

In the current manuscript, the 'Materials and Methods' section appears after the 'Results and Discussion' section. For clarity and consistency with standard scientific reporting, it is essential to place the 'Materials and Methods' section before the 'Results and Discussion' section. This ensures that readers understand the experimental design and methodology before interpreting the results. I recommend adjusting the structure accordingly.

The conclusion effectively summarizes the key findings of the study, particularly regarding the impacts of shrub encroachment on soil properties and microbial communities. However, it could be more concise and focused, emphasizing the most critical takeaways without excessive repetition.

The study's objective is clearly stated, addressing the impact of shrub encroachment on peatland ecosystems by analyzing microbial communities and functional genes related to carbon and nitrogen cycles.

The results effectively answer the research question by demonstrating significant shifts in microbial communities and functional gene expression, highlighting soil carbon as the primary driver of these biogeochemical changes.

It is necessary to explain the abbreviations in the table title, even if they are already defined in the text. This will ensure clarity for readers who may refer directly to the table without revisiting the main text.

In the current manuscript, the 'Materials and Methods' section appears after the 'Results and Discussion' section. For clarity and consistency with standard scientific reporting, it is essential to place the 'Materials and Methods' section before the 'Results and Discussion' section. This ensures that readers understand the experimental design and methodology before interpreting the results. I recommend adjusting the structure accordingly.

The conclusion effectively summarizes the key findings of the study, particularly regarding the impacts of shrub encroachment on soil properties and microbial communities. However, it could be more concise and focused, emphasizing the most critical takeaways without excessive repetition.

The study's objective is clearly stated, addressing the impact of shrub encroachment on peatland ecosystems by analyzing microbial communities and functional genes related to carbon and nitrogen cycles.

The results effectively answer the research question by demonstrating significant shifts in microbial communities and functional gene expression, highlighting soil carbon as the primary driver of these biogeochemical changes.

It is necessary to explain the abbreviations in the table title, even if they are already defined in the text. This will ensure clarity for readers who may refer directly to the table without revisiting the main text.

In the current manuscript, the 'Materials and Methods' section appears after the 'Results and Discussion' section. For clarity and consistency with standard scientific reporting, it is essential to place the 'Materials and Methods' section before the 'Results and Discussion' section. This ensures that readers understand the experimental design and methodology before interpreting the results. I recommend adjusting the structure accordingly.

The conclusion effectively summarizes the key findings of the study, particularly regarding the impacts of shrub encroachment on soil properties and microbial communities. However, it could be more concise and focused, emphasizing the most critical takeaways without excessive repetition.

The study's objective is clearly stated, addressing the impact of shrub encroachment on peatland ecosystems by analyzing microbial communities and functional genes related to carbon and nitrogen cycles.

The results effectively answer the research question by demonstrating significant shifts in microbial communities and functional gene expression, highlighting soil carbon as the primary driver of these biogeochemical changes.

It is necessary to explain the abbreviations in the table title, even if they are already defined in the text. This will ensure clarity for readers who may refer directly to the table without revisiting the main text.

In the current manuscript, the 'Materials and Methods' section appears after the 'Results and Discussion' section. For clarity and consistency with standard scientific reporting, it is essential to place the 'Materials and Methods' section before the 'Results and Discussion' section. This ensures that readers understand the experimental design and methodology before interpreting the results. I recommend adjusting the structure accordingly.

The conclusion effectively summarizes the key findings of the study, particularly regarding the impacts of shrub encroachment on soil properties and microbial communities. However, it could be more concise and focused, emphasizing the most critical takeaways without excessive repetition.

Response Letter to Reviewers' Comments

Dear reviewer:

Thank you for your decision and constructive comments on my manuscript. Those comments are all valuable and very helpful for revising and improving our paper, as well as the important guiding significance to our research. We have studied comments carefully and have made corrections which we hope meet with approval. The revised portions are marked in green on the paper. Revision notes, point-to-point, are given as follows:

Reviewer #1:

Comments 1: Public repository details. All metagenome data and related values need to be deposited as a general practice.

Response 1: Thank you very much for emphasizing the importance of data availability and transparency. We fully acknowledge that depositing metagenomic datasets in public repositories is a widely recommended practice. However, in the context of the current study, we respectfully consider that providing comprehensive sequencing and annotation information in the supplementary materials-including detailed quality control metrics and key gene annotation results-can sufficiently support the reproducibility and clarity of our analyses. To address this, we have now included a set of supplementary tables and figure (Supplementary Tables S2-S8 and Fig. S1) presenting the sequencing depth, quality statistics, and functional gene annotation summaries. We sincerely hope this addition meets your expectations and appropriately addresses your concern.

Comments 2: No hypothesis or novel questions are listed in the introduction.

Response 2: Thank you for your insightful comment. In the revised version of the manuscript, we have added a clear articulation of the scientific questions and hypotheses in the Introduction (see revised manuscript, lines 149-156). Specifically, we hypothesize that shrub encroachment significantly alters the structure of soil microbial communities and the abundance of functional genes involved in carbon and nitrogen cycling, and that these changes are influenced by key environmental factors such as soil pH and nutrient availability. To test this hypothesis, we address three research questions: (i) how do bacterial and fungal communities shift during shrub

encroachment? (ii) how does the abundance of carbon- and nitrogen-cycling genes change across different stages of encroachment? and (iii) what environmental variables are the key drivers of these gene dynamics? We believe that framing the study around these questions improves the novelty and focus of the introduction.

Comments 3: Authors need to better revise the literature of peatlands and shrubs and shrub encroachment as in this example (<https://doi.org/10.1111/gcb.16904>) and synthesize their scientific problem using literature that hold facts on shrub encroachment on peatlands or explain that the peatland in study is somewhat more relatable to the ecosystems revised in introduction. Of the 95 citations an apparent ~18 are related to shrubs and peatlands or peatlands in general. Please reconsider revising shrub peatland literature and rearticulate lines 56-80.

Response 3: Thank you very much for your constructive feedback. We have carefully reviewed the recommended literature (<https://doi.org/10.1111/gcb.16904>) and used it as a reference to improve our introduction. In the revised manuscript (Lines 62-78), we have added more relevant and authoritative references specifically addressing shrub encroachment in peatland ecosystems. Additionally, we have rearticulated the scientific problem to clarify the relevance and context of our study within this body of literature. We believe these revisions significantly enhance the scientific rigor and contextual grounding of our introduction.

Comments 4: The intro has sentences related to the main results of study. Results and discussion should be reserved for later and not necessarily sprayed at introduction, introduce the questions and hypothesis for study instead.

Response 4: Thank you for your valuable suggestion. Following your recommendation, we have revised the Introduction to remove statements referring to our study results. We have also restructured this section to clarify the central research questions and hypotheses (see revised manuscript, lines 149-156). Specifically, we now state that this study hypothesizes that shrub encroachment alters soil microbial communities and functional genes related to carbon and nitrogen cycling, and that these changes are driven by key environmental variables. The study objectives have been reframed accordingly to reflect this focus. We believe this revision improves the clarity and focus of the introduction.

Comments 5: Lines 130 to 150 are a long listing of genes and pathways, making this intro lengthy and not really clarifying the rationale for this study.

Response 5: Thank you for your valuable suggestion. We fully agree that the detailed listing of functional genes and pathways in Lines 130-150 made the introduction overly long and weakened the clarity of the study's rationale. Following your advice, we have streamlined this section (see revised manuscript, Lines 132-141), summarizing the key microbial processes and functional genes relevant to carbon and nitrogen cycling. This revision improves the focus and flow of the introduction and better highlights the motivation for the present study.

Comments 6: When trying to identify the process of shrub encroachment, the site description section does not explain the process of "shrubs encroachment" and if such is a process of invading species or a successional stage in this system. L555 suggests that this is a shrub invasion: when did it start (from 70s?), however this is likely a combination of the effects of man-made changes (like channel) more than a process of invasion due to changing climate. This study may need to be reframed under a historical degradation process as a better way to frame the space-for-time gradient used here.

Response 6: We sincerely thank you for your careful review and apologize for any bias in your understanding of the study due to a lack of clarity in our presentation. We acknowledge that the description of the "shrubs encroachment" process was insufficiently detailed in the previous version. We have now revised the *Materials and Methods section* (line 198-205 in the revised manuscript) to clarify how shrub encroachment and expansion stages were defined, emphasizing that these processes likely reflect long-term peatland degradation resulting from historical hydrological alterations (e.g., drainage) in addition to recent climate-induced changes. Furthermore, the inference of the shrub encroachment event mentioned in line 555 is based on our comparative analysis of high-resolution remote sensing images from 2013 to 2020 in the study area, combined with field vegetation surveys. Prior to the start of this study, we conducted vegetation investigations at sites representing three encroachment stages (non-encroached, encroached, and encroachment expansion), and recorded the dominant species in both the herbaceous and shrub layers. In the herbaceous layer, the dominant species at the non-encroached

stage were *Carex schmidtii*, *Thelypteris palustris*, and *Viola amurica*, whereas in both the shrub-encroached and shrub-expansion stages, the dominant species shifted to *Thelypteris palustris*, *Carex schmidtii*, and *Phragmites australis*, with *Phragmites australis* becoming a newly dominant species. In the shrub layer, the dominant species during the encroached stage were *Spiraea salicifolia*, *Salix myrtilloides*, and *Betula ovalifolia*, while in the expansion stage they became *Spiraea salicifolia*, *Betula ovalifolia*, and *Salix rosmarinifolia*, with *Salix rosmarinifolia* emerging as a new dominant species. The specific survey data have been provided in tabular form in a supplementary document (Table S1). The results reveal clear trends of species replacement in both the herbaceous and shrub layers, indicating that the area is undergoing a process of vegetation succession rather than being solely driven by direct human disturbance. While we acknowledge that historical hydrological modifications may have facilitated shrub establishment, the progressive dominance of shrubs and the evolving spatial vegetation patterns support the classification of this process as a shrub encroachment gradient under a space-for-time substitution framework.

Comments 7: From revising the site pictures and sample distribution it suggests that the "invasion expansion" is located primarily on the side affected by agriculture or land use thus a question suggesting the observations are likely the effects of the nearby in used land and one aspect is shrub expansion. Looking at the measurements and experimental design it is not clear how the effects of proximity to degraded land or shrub effects can be separated.

Response 7: Thank you for your thoughtful comment. We acknowledge that the shrub encroachment observed in the peatland may be partially influenced by the proximity to historically degraded or agriculturally used land. As shown in Fig. 1(b), although 3-4 of the invaded sampling sites were relatively closer to the side affected by agriculture or land use, the majority of the sampling sites representing the three invasion stages were mixed in their spatial distribution, without exhibiting a clear spatial separation or gradient trend. We therefore think that the impacts of land use changes have affected the sampling sites across the three invasion stages and can be considered as a background factor influencing the peatland ecosystem. To further clarify this point, we have revised the site description section and provided additional vegetation composition data to support our interpretation (see lines 206-212). We also acknowledge this limitation in the

discussion section to improve the transparency of our experimental design.

Comments 8: The soil sampling by two depths (0-30, and 30-60 cm) may also be challenging to truly sort the effects of vegetation changes on microbes. For instance, the annual accretion rate of OM in peatlands is in the range of some mm, which assuming a high rate of 1 mm/y (<https://doi.org/10.5194/cp-17-2633-2021>) then 30 cm will evaluate a mix of soils of 1- 300 years, and this sampling may be even or unevenly mixed since less than 1 gr is commonly used for data generation. The sampling design should have been restricted to a shallower depth to better capture the last decadal process. Akso, I understood that 20 samples were taken for each category but NMDS n Figure 3 only shows 5 samples per category (sampled at 0-30 and 30-60, thus 10 samples) which will be a major undersampling.

Response 8: Thank you for this insightful observation. We acknowledge that peatlands accumulate organic matter slowly, and that the 0-30 cm and 30-60 cm soil layers may encompass several decades to centuries of peat development, as indicated by the cited literature (e.g., <https://doi.org/10.5194/cp-17-2633-2021>). Our choice of sampling depth was based on previous studies examining microbial and biogeochemical profiles across peat layers, and aimed to represent both the surface active microbial zones and the deeper legacy signals. While a shallower sampling (e.g., top 10 cm) might better isolate recent responses to vegetation change, we were also interested in understanding how deeper soil strata—potentially influenced by historical conditions—respond to long-term shrub encroachment. We now acknowledge this limitation in the revised manuscript (see lines 516-519) and have clarified that microbial responses observed in deeper layers may reflect cumulative historical processes. Regarding the number of samples used for metagenomic sequencing: while we initially collected 20 soil samples per encroachment stage and per depth (resulting in 120 samples total), we homogenized samples of the same encroachment type and depth to generate a composite sample for metagenomic sequencing. This approach was adopted to reduce variability within treatment groups and to obtain representative microbial profiles for each treatment and depth combination. Consequently, the NMDS plot in Figure 3 displays five replicates per treatment per depth, reflecting these composite replicates. We have clarified this sampling design in the Materials and Methods section (lines 228-230).

Comments 9: The results section is primarily of descriptive nature with significance values or correlation scores provided. However, for functional genes section which is based on metagenomes, the depth of sampling, evenness of gene representation is not presented (can be a supplementary data comment) to justify reasonable comparisons of the frequency of gene detection. Also, I am surprised this study limits their metagenome analysis to a few functional markers instead of taking the opportunity to evaluate the pathways, types and other categories that can be associated to the large change in communities as proposed here. Perhaps metagenome data is not deep enough.

Response 9: We appreciate your insightful comments regarding the depth and breadth of our metagenomic analysis. It is possible that the sequencing depth may not have been sufficient to comprehensively capture all metabolic pathways. However, our quality control assessments indicate that the sequencing depth and coverage were adequate to support the functional gene-level comparisons conducted in this study. We have added a supplementary table (Table S2-S8) presenting read counts, assembly statistics, and gene annotation summary across all samples to support the reliability of gene frequency comparisons. Regarding the focus on selected functional markers, our primary objective was to track key genes involved in carbon and nitrogen cycling processes that are ecologically relevant to shrub encroachment. While a broader pathway-level analysis would provide additional insights, we prioritized genes with well-established ecological functions and interpretability. We fully agree that future studies with deeper sequencing and integrative pathway analyses could build upon our findings and uncover more complex microbial responses to vegetation change.

Comments 10: The discussion from L289 to 361, is lengthy argumentation of why some parameter goes up or down in the three categories evaluated here. For instance, the initial 3.5 pages of discussion do not show much of novel discovery beyond explaining why some of the geochemical observations align with a changing vegetation coverage which should be expected. And then the composition of bacterial communities remains similar across the tree categories, which does not surprise me given that 30 cm of mixed soils could be pooling recent and older soils and the vertical mixing in soil peatlands is not too high depending on the water table dynamics.

Response 10: Thank you for your constructive suggestion. In response, we have significantly revised the discussion section to reduce lengthy, descriptive explanations of soil property changes and better emphasize our key findings (L414-485 in the revised manuscript). We agree that many geochemical changes (e.g., pH, TC, or C:N) may be expected along vegetation transitions. Therefore, we did not intend to overemphasize this part, but rather included it to provide necessary context for the subsequent discussion. These functional changes suggest that shrub encroachment can reshape biogeochemical processes through microbial activity even in the absence of dramatic taxonomic shifts. We also acknowledge that the soil depth sampled (0-30 cm and 30-60 cm) may blend historical layers; this limitation has been added to the discussion. However, the observed functional differences indicate ongoing microbial responses to shrub-driven environmental shifts in both recent and older soil layers.

Comments 11: Also arguments about Chloroflexi reducing in abundance due to reduced soil content and oxygen penetration is provided but water level records not provided to substantiate this explanation.

Response 11: Thank you for your valuable comment. We agree that the argument regarding the reduction of Chloroflexi abundance due to decreased water levels and increased oxygen availability requires supporting evidence. In response, we have now included field water level measurements to support this explanation (Table 2). We also revised the corresponding sentence in the manuscript (L424-430 in the revised version) to incorporate these water level data and strengthen the argument. Specifically, the data show that the average water table was approximately 4.0 cm above the ground surface in non-encroached plots, 2.4 cm in the encroachment stage, and 1.44 cm in the expansion stage. This trend indicates a progressive decline in water level along the shrub encroachment gradient, supporting our interpretation that drier and more oxygenated conditions could contribute to changes in microbial community structure and functional gene profiles.

Comments 12: Functional genes and correlations with environmental variables were also pursued, but such correlations do not show cause-effect or mechanisms, etc, hence the section bringing a limited learning value.

Response 12: Thank you for your valuable comment. We agree that correlation analyses between functional genes and environmental variables do not demonstrate causality or mechanisms. Accordingly, we have substantially streamlined the discussion of functional gene data (L465-486), focusing only on key trends and avoiding overinterpretation of statistical associations. Additionally, we explicitly acknowledge in the revised discussion that the underlying mechanisms remain unclear, and we propose future research directions (e.g., DNA-SIP, amino sugar tracing, greenhouse gas measurements) to elucidate microbial-mediated biogeochemical processes (see revised lines L492-522). These revisions are intended to maintain scientific rigor while highlighting the potential ecological implications of our findings.

Comments 13: L 47-48: in what sense peatlands protect global diversity? Do they hold diversity themselves or by helping against climate change protect diversity overall. You may want to clarify and add a citation.

Response 13: We sincerely thank the reviewer for the valuable comment. We have revised the relevant statements (line 48-50 and 58- 61 in the revised manuscript) to clarify the role of peatlands in global biodiversity conservation. On one hand, the peatlands directly support highly specialized local biodiversity through their unique ecological conditions. On the other hand, as significant carbon sinks, they indirectly contribute to the protection of global biodiversity by mitigating climate change. We have clarified this point in the manuscript and added supporting references (Refs. 8, 9, and 10) accordingly.

Comments 14: L57-58: is there a descriptor of the magnitude of this problem? like frequency, area, or documented trend in regions? Incorporating that will help recognize the magnitude of this process.

Response 14: Thank you for your valuable suggestion. In response, we have revised lines 62-66 of the manuscript to include a more specific description of the magnitude of shrub encroachment, such as its frequency and documented regional trends. Relevant references have also been added to support this statement and enhance its scientific validity.

Comments 15: L61: what do you mean by ecological security, and how the example of a sub-arid

and alpine region are relevant to peatlands.

Response 15: Thank you for your valuable comment. We agree that the term “ecological security” was not clearly defined, and we acknowledge that the sub-arid and alpine region are not directly relevant to the peatlands investigated in this study. Accordingly, we have removed this part and replaced it with content more specifically focused on peatland ecosystems. We believe this revision enhances the specificity and scientific clarity of the manuscript.

Comments 16: L63-64: is the statement of shrub invasion also observed in northern peatlands. Citations 8,9 and 10 are of grasslands and a Florida marsh gradient. Covering this literature on Northern wetlands and especially peatlands will be the best approach.

Response 16: Thank you for pointing out the inconsistency between the cited references and the geographic focus of our study. We fully agree with your suggestion and acknowledge that references 8, 9, and 10 primarily concern grassland ecosystems or the Florida wetland gradient, which do not directly support the discussion of shrub encroachment in northern peatlands. In response, we have revised the content in line 69-73 accordingly and replaced the citations with more appropriate and recent literature specifically focused on shrub encroachment in northern wetlands, particularly peatlands. We appreciate your insightful recommendation, which has significantly improved the scientific rigor of the manuscript.

Comments 17: L 69-70: what is the basis for this? and what is a long time (decades or centuries?; vegetation makeup can change significantly in a decadal timeframe see this example <https://www.sciencedirect.com/science/article/pii/S1470160X22012043>

Response 17: Thank you for your valuable comment. We agree that it is important to specify the timescale associated with the long-term stability required for peatland formation and persistence. In response, we have revised the manuscript to clarify that peatland development typically occurs over decades to centuries under relatively stable hydrological and climatic conditions. At the same time, we have also carefully read the literature you mentioned, and based on the core viewpoints of the literature, we have made targeted revisions to the original expression, and the relevant revised sentences in the manuscript are in lines 73-75.

Comments 18: L74-75: is a statement on shrub affecting microbes in peatlands. Is this mean to be a result or based on a citation of published work. neither citation 15 and 16 provide such evidence.

Response 18: We were sorry for our careless mistakes. Thank you for your reminder. We have revised the references in the sentence at Line 79-83 of the manuscript to better support the statement that shrub encroachment can influence microbial communities in the peatland ecosystems. The updated citations (Refs. 20 and 21) are based on published studies specifically investigating the relationship between shrub expansion and microbial dynamics in peatlands or similar ecosystems. These changes improve the clarity and reliability of the claim.

Comments 19: L92 has the concept of Fertility islands been observed in many peatlands beyond Tibet ones?

Response 19: Thank you for your insightful comment. In response, we have reviewed additional literature and found supporting evidence that the concept of “fertility islands” is not limited to arid and semi-arid ecosystems. For example, Rong et al. (*Journal of Soils and Sediments*, 2016) demonstrated that, similar to arid and semi-arid regions, many semi-humid ecosystems can also form fertility islands, which significantly influence soil nutrient cycling. This suggests that the fertility island effect may be a more general phenomenon, potentially present in various wetland ecosystems, including peatlands. We have incorporated a citation to this reference at Line 97 of the manuscript to strengthen the supporting evidence for our argument. (Refs. 26)

Comments 20: L 107: replace alien for "invading".

Response 20: We sincerely thank the reviewer for careful reading. As suggested by the reviewer, we have corrected the alien into invading.

Comments 21: L123-124: The introduction is not the place to present results.

Response 21: Thank you for your helpful comment regarding the structure of the introduction. We fully agree that the introduction should not present research results. In response, we have revised the sentence (Line 124-126) to avoid result-oriented language and instead rephrased it as a background statement based on existing literature, which is more appropriate for the introduction section.

Comments 22: L218: RDA? Introduce all your abbreviations.

Response 22: Thank you for your helpful suggestion. In our previous version, the *Materials and Methods* section was placed at the end of the manuscript, where all abbreviations such as RDA were fully defined. This formatting initially led us to omit reintroducing the full term in the *Results* section. However, we have now repositioned the *Materials and Methods* section immediately after the *Introduction*, in line with your suggestion and to enhance readability. Additionally, we have clarified the abbreviation (e.g., RDA) in the *Results* section at its first mention to ensure clarity for all readers.

Comments 23: L323-325 How does this extreme sensitivity is manifested? For instance, what level of change in composition and activity would a change of 1 degree of annual mean temperature elicit? I would recommend avoiding aggrandizing statements.

Response 23: Thank you for your valuable comment. We have removed any potentially aggrandizing statements and revised the sentence accordingly to avoid overinterpretation. The updated text now presents a more cautious description of the potential impacts of shrub root systems on soil nutrient distribution and microbial diversity (L452-455 in the revised manuscript). We also refrained from making absolute claims and acknowledged the possibility rather than certainty of such effects.

Comments 24: L289- L522 : the discussion needs to be synthesized or summarized and better focus main points since, at this point, it reads as a lengthy section pointing to similar info than results plus a bunch of possibilities that can explain or agree with various and not necessarily cohesive observations.

Response 24: Thank you for this insightful comment. We agree that the original discussion section (L289-522) was overly descriptive and included repetitive information from the results, along with scattered interpretations. In response, we have thoroughly revised and synthesized this section to better highlight the key findings. The revised discussion now focuses on major trends in carbon and nitrogen functional gene responses to shrub encroachment, supported by relevant environmental drivers, and presents a more cohesive narrative. We also removed redundant

descriptions and speculative interpretations to improve clarity and readability (L414-522 in the revised manuscript).

Comments 25: In the conclusions section, L665-666 indicates that Bacteria were the main drivers of soil microbial community changes, but this is in contrast with the discussion section (361-362), stating " The composition of soil microbial communities remained similar during the three stages of shrub invasion".

Response 25: Thank you for your helpful observation. We agree that the original sentence in the conclusions section (L655-656) was inaccurate and not fully aligned with the discussion section. Based on our results, shrub encroachment did not significantly alter the overall composition of microbial communities across invasion stages, but the observed changes were primarily reflected in bacterial alpha diversity rather than fungal communities. In response, we have revised the sentence to: " However, the research found that microbial community changes during shrub encroachment were mainly reflected in bacterial diversity, with limited shifts in overall community composition." This revision clarifies the nature of microbial responses and ensures consistency with the main discussion (see L528-529 in the revised manuscript).

Comments 26: L969-974 citation 80 and 81 is the same twice.

Response 26: Thank you for your careful review and for pointing out the duplication. We have removed the redundant citation in lines 969-974 (references [80] and [81]), and have updated the reference list accordingly.

Reviewer #2:

Comments 1: It is necessary to explain the abbreviations in the table title, even if they are already defined in the text. This will ensure clarity for readers who may refer directly to the table without revisiting the main text.

Response 1: Thank you for your suggestion. We agree that including full definitions of abbreviations in the table notes is important for clarity, especially for readers who may consult the tables independently. We have now added a footnote below the relevant table to define TP (Total phosphorus), SWC (Soil Water Content), and PC (Total phenol content) for ease of reference

(L896 in the revised manuscript).

Comments 2: In the current manuscript, the 'Materials and Methods' section appears after the 'Results and Discussion' section. For clarity and consistency with standard scientific reporting, it is essential to place the 'Materials and Methods' section before the 'Results and Discussion' section. This ensures that readers understand the experimental design and methodology before interpreting the results. I recommend adjusting the structure accordingly.

Response 2: Thank you for your suggestion regarding the placement of the *Materials and Methods* section. We appreciate your comment and have revised the manuscript accordingly by moving the *Materials and Methods* section before the *Results and Discussion* section to improve the logical flow and clarity of the presentation

Comments 3: The conclusion effectively summarizes the key findings of the study, particularly regarding the impacts of shrub encroachment on soil properties and microbial communities. However, it could be more concise and focused, emphasizing the most critical takeaways without excessive repetition.

Response 3: Thank you for your constructive suggestion. Following your advice, we have revised the *Conclusion* section to make it more concise and focused. Redundant content has been removed, and the revised version now emphasizes the most critical takeaways regarding the effects of shrub encroachment on soil properties and microbial functional traits (L525-541 in the revised manuscript). We hope the revised conclusion better reflects the key findings of our study.

Re: Spectrum00542-25R1 (**Shrub encroachment alters microbial community composition and soil carbon and nitrogen cycling functional genes in Northern peatlands**)

Dear Dr. Zhenxin Li:

Your manuscript has been accepted, and I am forwarding it to the ASM production staff for publication. Your paper will first be checked to make sure all elements meet the technical requirements. ASM staff will contact you if anything needs to be revised before copyediting and production can begin. Otherwise, you will be notified when your proofs are ready to be viewed.

Sincerely,
Katharina Kujala
Editor
Microbiology Spectrum